# Accurate Prediction of Transcriptional Activity of Single Missense Variants in HIV Tat with Deep Learning

**DOI:** 10.3390/ijms24076138

**Published:** 2023-03-24

**Authors:** Houssemeddine Derbel, Christopher J. Giacoletto, Ronald Benjamin, Gordon Chen, Martin R. Schiller, Qian Liu

**Affiliations:** 1Nevada Institute of Personalized Medicine, University of Nevada Las Vegas, 4505 S Maryland Pkwy, Las Vegas, NV 89154, USA; 2School of Life Sciences, College of Sciences, University of Nevada Las Vegas, 4505 S Maryland Pkwy, Las Vegas, NV 89154, USA

**Keywords:** variant effect, deep learning, HIV, tat protein

## Abstract

Tat is an essential gene for increasing the transcription of all HIV genes, and affects HIV replication, HIV exit from latency, and AIDS progression. The Tat gene frequently mutates in vivo and produces variants with diverse activities, contributing to HIV viral heterogeneity as well as drug-resistant clones. Thus, identifying the transcriptional activities of Tat variants will help to better understand AIDS pathology and treatment. We recently reported the missense mutation landscape of all single amino acid Tat variants. In these experiments, a fraction of double missense alleles exhibited intragenic epistasis. However, it is too time-consuming and costly to determine the effect of the variants for all double mutant alleles through experiments. Therefore, we propose a combined GigaAssay/deep learning approach. As a first step to determine activity landscapes for complex variants, we evaluated a deep learning framework using previously reported GigaAssay experiments to predict how transcription activity is affected by Tat variants with single missense substitutions. Our approach achieved a 0.94 Pearson correlation coefficient when comparing the predicted to experimental activities. This hybrid approach can be extensible to more complex Tat alleles for a better understanding of the genetic control of HIV genome transcription.

## 1. Introduction

Human immunodeficiency virus (HIV) causes acquired immunodeficiency syndrome (AIDS), characterized by a progressive failure of the immune system. It remains an important health problem in the United States with 1,189,700 infected people, 18,489 annual deaths, and an annual medical cost exceeding USD 50 billion [1]. HIV lacks proofreading of its replicated RNA genome and has a high mutation rate of 1 in 10^4^ bp, with each virion 9 kB genome having about 10 new variants [2]. Furthermore, an active HIV infection in a single individual is estimated to generate approximately 10^11^ virions per day [3]. The combination of high mutation rates with efficient virion generation creates an extremely genetically heterogeneous and diverse viral genome population, which is a key consideration for important pathogenic processes such as antiretroviral therapy (ARV) resistance, latency, and strain evolution. After selective pressure from ARV therapy, variant virions with drug-resistant variants may survive and propagate, limiting therapeutic efficacy. Therefore, it is important to understand how HIV evolves both within a person, and in worldwide populations with relevance to AIDS pathogenesis and treatment [4].

Tat is an essential regulatory gene that drastically enhances the efficiency of HIV genome transcription and replication. The absence of Tat may lead to short and abortive viral transcripts, and Tat variants widely affect different viral activities. Therefore, a comprehensive investigation of various activities of Tat variants can deepen the understanding of AIDS pathology and assist drug design targeting for a broader range of HIV-1 strains.

Studies of variant frequencies, HIV evolution, and small-scale mutagenesis studies have greatly advanced knowledge about ARV drug resistance and how to effectively treat AIDS. Although viral isolates from an infected patient are experimentally tested, the ability to link specific genetic changes with the functions of viral proteins is largely limited to low-throughput experiments, which slowly and incrementally reveal how the vast variant landscape of a typical infection impacts HIV replication, viral latency, drug resistance, and AIDS pathogenesis.

Two high-throughput approaches are available to estimate the effects of the variants: the GigaAssay [5] directly measures a functional readout such as transcription, whereas the alternative multiplex assays of variant effect (MAVEs) are survival screens [6,7,8]. Activities are determined in a GigaAssay by measuring thousands of reads for approximately a million individually UMI-barcoded variant cDNAs. By comparing populations of cDNAs for each mutant to the populations for controls, this GigaAssay approach produces an accurate measurement and classification of Tat transcriptional activity with high confidence.

A previous analysis of Tat with the GigaAssay reported transcriptional activities for all 1615 Tat single and 3429 double missense variants with a ~95% accuracy [5]. A total of 35% of all possible single amino acid variants in Tat are loss-of-function. However, it is currently too time-consuming and costly to conduct GigaAssay experiments on millions of variants to complete the Tat double missense mutant landscape. Our aim is to engineer efficient computational approaches to accurately estimate Tat’s transcriptional activities of single variants. These tools will next be used in a future extension to predict the activities of double-variant Tat alleles. The novelty of our approach lies in the combination of high-accuracy GigaAssay variant/activity data with deep learning algorithms. The evaluations herein demonstrate the efficiency of our novel approach in predicting single missense mutant activities. This clearly suggests that this approach can likely be extended to predict the effect of more complex variants and possibly for other protein activities.

## 2. Results

### 2.1. Overview of the Proposed Deep Learning Framework Called Rep2Mut

We proposed testing a deep learning framework called Rep2Mut to accurately estimate the transcriptional activity of missense variants. The architecture of the Rep2Mut algorithm is shown in Figure 1 using the Tat protein (86 amino acids) as an example. The output of Rep2Mut is the predicted effect upon the transcriptional activity of Tat variants, and the inputs are the wild type (WT) protein sequence, mutated protein sequences (missense variants), and mutated amino acid positions.

There are several steps in the Rep2Mut algorithm to capture the difference between a WT sequence and its mutated sequences. First, a WT sequence is used as an input of the evolutionary scale modeling (ESM) protein language model [9] to learn the representation of the position of interest in the WT sequence. This WT representation, with a vector of 1280 elements, is then an input of fully connected layer 1 (Figure 1) to generate a vector with 128 elements. Similarly, the corresponding mutated sequence is used as input to the ESM to generate a representation vector of 1280 elements in the mutated sequence. The learned representation is then fed into a fully connected layer 2 (Figure 1) to generate the other vector with 128 elements. The two vectors of 128 elements are combined by applying element wise multiplication (see Figure 1) followed by concatenation with the position encoding vector of a mutation position. The position encoding vector has N elements (N = 86 for the Tat protein); each element is for one position in the protein sequence. All values are zero except for the mutated position, which is marked by 1. Next, the combined vector is an input to fully connected layer 3 (Figure 1) with a Sigmoid activation function to generate the prediction of transcriptional activity. In total, the Rep2Mut network with the Tat protein has 328,153 trainable weights.

### 2.2. Evaluation of the Proposed Deep Learning Framework Rep2Mut

We evaluated Rep2Mut on the GigaAssay transcriptional activity data for all 1615 single amino acid missense variants in HIV Tat. Layers 1 and 2 of Rep2Mut (in Figure 1) were pretrained on 115,997 single variants of 37 existing protein datasets with different protein functional measurements (Figure 1). After adding layer 3, all layers in Rep2Mut were then optimized and fine-tuned on the experimental GigaAssay data. To avoid overfitting, 10-fold cross-validation was repeated 10 times and used to calculate the performance of Rep2Mut.

When the variant activities predicted with Rep2Mut were compared to the GigaAssay results, a Pearson correlation coefficient of 0.94 and a Spearman correlation coefficient of 0.89 were observed. We repeated the analysis with a baseline method and two recently published methods, ESM [9] and DeepSequence [10], compared to Rep2Mut in Figure 2 and Table 1. The baseline method is a feed forward neural network with simple encoding of the variant sequences as input (described in the Methods), and its performance was ~0.17 lower than Rep2Mut.

For the ESM prediction methods (called ESM_pred, to be distinguished from ESM models), we tested all Tat variant activities of the five trained ESM_pred estimations compared with the experimental activities and calculated the best performance. The predictions and the performance of the averaged prediction by ESM_pred are shown in Figure 2a,b and in Table 1. As expected, the averaged estimation of variants from the five ESM_pred achieved a better performance (0.59 Spearman correlation coefficient) than any of the individual ESM_pred estimations.

For the DeepSequence method, we generated multiple sequence alignments using EVision and retrained DeepSequence as suggested by Riesselman et al. [10]. DeepSequence needs to be retrained for each protein sequence due to the different number of dimensions. DeepSequence generated a higher Pearson correlation coefficient (0.57), but lower Spearman correlation coefficient (0.41) when compared to the ESM_pred prediction of the activities of the Tat variants (Figure 2c and Table 1). Unfortunately, DeepSequence was only able to generate predictions for 10% of all variant data (see Figure 2c), even after fine-tuning the retraining process with more sequences in the multiple sequence alignments, demonstrating a limitation of DeepSequence for this application.

In conclusion, our Rep2Mut algorithm achieved a much better performance when compared to the state-of-the-art models ESM_pred and DeepSequence (Figure 2d) and (Table 1). The Pearson correlation coefficient for Rep2Mut was 0.39 higher than ESM_pred, 0.37 higher than DeepSequence, and 0.18 higher than the baseline method. Likewise, the Spearman correlation coefficient was 0.31 higher than ESM_pred, 0.48 higher than DeepSequence and 0.17 higher than the baseline method.

### 2.3. Effect of Amino Acid Position on Activity Prediction

The activities of Tat variants are partially dependent upon the positions in the Tat protein sequence. Figure 3 shows that the last 20 positions of the Tat protein (C-terminal) have WT GigaAssay activities with some outliers such as K85E, which demonstrate a higher tolerance than the N-terminal amino acids [5]. Therefore, we tested the predictions of the Rep2Mut without a position encoding vector. Compared with Rep2Mut, a modified algorithm without a position vector achieved slightly lower Pearson and Spearman correlation coefficients (0.03 and 0.02 lower, respectively; Table 1). This result suggests that Rep2Mut has no significant overfitting with positional information.

### 2.4. Rep2Mut Sensitivity Analysis for the Fraction of Training Data

The initial evaluation of Rep2Mut predictions used 90% of the variant activity data (n = 1457) to train Rep2Mut, and the remaining 10% for testing. However, for scaling, even with high-throughput wet-lab experiments such as the GigaAssay, acquiring experimental data is too time consuming and cost prohibitive to assess complex variants. Therefore, we evaluated Rep2Mut’s performance to identify the minimal amount of training data needed to maintain near-maximal performance. We trained Rep2Mut with 70, 50, 30, 20, 10, and 7% of the Tat activity single missense variant dataset, and tested Rep2Mut performance with the remainder of the data. To reduce errors from random sampling, we split, trained, and tested those predictions 50 times, calculating an average performance for all tests (Figure 4).

As expected, reduced Rep2Mut performance was observed with smaller training sampling. However, the performance (Spearman correlation coefficient) only showed a minimal reduction when Rep2Mut was trained with 50% or more of the data. Further reduction to 20 or 30% of the training data only reduced the performance by 0.03. The performance further decreased by another ~0.02 when 10% of the data was used for training. Surprisingly, Rep2Mut achieved more than a 0.80 or 0.82 Spearman correlation coefficient when only 7% (113 variants) or 10% of the data, respectively, were used for training. This will make the combination of deep learning with the GigaAssay more scalable for multivariant alleles because as little as ~20% of the variant data can generate predictions of variant effect with little compromise in performance.

## 3. Discussion

### 3.1. Visualization of Predicted Vectors

To better understand how Rep2Mut predicts variant effects, we used combined vector after dot product in Figure 1 of all variants to create a global map after dimension reduction to a 2D space using uniform manifold approximation and projection (UMAP) [11]. We then investigated the resulting 2D map for correlations with GigaAssay activity, variant position, and the physiochemical types of amino acids (Figure 5).

Figure 5a,b clearly demonstrate that variants with different experimental activities had a smooth distribution from the right to the left, with or without position encoding. Variants with high activities were in the left half, while most variants with low experimental activities were in the right half for both Figure 5a,b. The positions followed quite a similar distribution in the 2D space, although there was abnormal deviation in the middle of the plots of Figure 5c,d. This again suggests that the Rep2Mut model itself learned position information from the protein sequences without including the position encoding.

### 3.2. Association of Amino Acid Types with Tat Activity Predictions

Figure 5e–j show three different groups of WT amino acid types in 2D space, with and without the position vector. In all subplots, there was a clear distribution of experimental activities from right to left. Interestingly, there was no such pattern in the 2D space with different groups of mutated amino acids, suggesting non-randomness of the WT amino acids at each position.

### 3.3. Outliers in Rep2Mut Prediction

To better understand incorrect predictions, we analyzed the activity prediction outliers. In Figure 2d and Table 2, we annotated those variants whose predicted activities were 0.3 larger or smaller than the experimentally-determined activities (n = 20). We split them into two groups: overestimation if the predicted activities were 0.3 larger than the experimental activities, or underestimation if predicted activities were 0.3 smaller than experimental activities. We chose the 0.2 to 0.3 range because this is the approximate error rate for activities determined for UMI barcodes in the GigaAssay [5]. In Table 2, we also listed the mean, maximum, and minimum of the predicted and GigaAssay activities for the positions of the outlier predictions.

In all outlier predictions, the average predicted activities of each mutated position were very similar to the averaged experimental activities for that position. The majority of the outlier overestimations had very low experimental activities among the 19 variants for each position, while all of the underestimated outliers had the highest experimental activities. In particular, K12P and K85E, which were overestimated by Rep2Mut, had significantly lower experimental values when compared to other variants at the same positions (K85E: 0.16 vs. >0.72 for other K85 variants; K12P: 0.12 vs. >0.5 for other K12 variants). Several other variants such as P3R, P6K, K12P, and R7P all involved a proline substitution, suggesting that the unique nature of proline might not be captured by the deep learning algorithm.

Curiously, all overestimated and underestimated outliers were in the Cyclin T1 interaction site defined in a structure of the Tat:Cyclin T1 complex [12]. Visualization of this structure (PDB: 4OR5) with PyMOL identified M39K and F32H in two α-helices (27–32 and 34–42) of the Tat protein, and the majority of these mutated positions interacted with Cyclin T1 (Figure 6). According to the accessible surface area calculated by RDBePISA, many of the mutated positions had a buried accessible surface area >40 Å^2^ when the Tat protein binds with Cyclin T1. The structure analysis also demonstrates that the outliers Q35R, Q17V, and E2A have hydrogen bonds with Cyclin T1.

One potential explanation of the observation of the erroneous predictions for substitutions at the Cyclin T1 interface is that Cyclin T1 accessibility or interactions may differ in the cell lines used in the GigaAssay (the cells used for the GigaAssay express Cyclin T1 [13]). An alternative explanation could be that our predictions do not have data that considers the structural complexity of the protein–protein interaction in this region. Another possibility is that these amino acids are important for a presently unknown interaction.

## 4. Materials and Methods

### 4.1. Dataset

The dataset used for testing was made up of single variants of HIV Tat proteins, generated by GigaAssay [5]. This Tat protein is composed of 86 amino acids, and each position except the first amino acid is mutated individually to 19 other amino acids besides the WT amino acid. In total, there are 1615 single missense variants of the Tat protein. Each missense variant was sequenced with more than five barcodes, and the transcriptional activity of each variant was calculated by the GigaAssay. The effect of a single variant was estimated with a value ranging from 0 to 1, where the larger the value, the less effect of the single variant on the transcriptional activities of the Tat protein.

### 4.2. Rep2Mut Framework to Estimate Tat Variants’ Effect on Transcriptional Activities

Rep2Mut is a sequence-based prediction of the effect of variants on transcriptional activities measured by GigaAssay. As shown in Figure 1, the input of Rep2Mut includes three types of sequence information. One is the WT sequence, and the other is the mutated sequence with a substitution of an amino acid at a position of interest. For either the WT or mutated sequence, we used evolutionary scale modeling (ESM) [9] to learn the representation of the mutated position.

ESM [9,14] is a self-supervised learning framework that was trained on millions of protein sequences to learn multiple levels of protein knowledge from biochemical properties to evolutionary information. It is composed of multiple transformer layers and trained using the masked language modeling objective [15]. Usually, the learned representation at the 33^rd^ layer is used to predict the diverse functions of proteins. ESM-1v [9] is a 34-layer transformer trained on a UniRef90 dataset [16] with five released pretrained models. We used the first pretrained model, and fed WT or mutated sequences into it. We used the learned vector of the position of interest at the 33^rd^ layer to represent WT or variant information. This learned vector has 1280 elements.

Each of the learned representation vectors from ESM-1v was used as the input of a fully connected neural network layer with a vector of 128 elements as the output (as shown in Figure 1). The PReLU activation function [17] was applied to the layers with a dropout rate of 0.2 to avoid overfitting [18]. The two 128-deminsion vectors were then merged with an elementwise dot product. The element-wise product (or the Hadamard product) is a binary operation that takes two matrices of the same dimensions as the input and produces another matrix of the same dimension as the operands. In other words, given two matrices *A_m,n_* and *B_m,n_*, of the same dimension *m* × *n*, the elementwise product $A⊙B={\left(A\right)}_{ij}{\left(B\right)}_{ij}$, where 0<i≤m and 0<j≤n.

The third type of input to Rep2Mut was a mutated position. A position was encoded into a binary vector of N elements each of which corresponds to a position in the protein sequence of interest. This encoding vector only has one value of 1 at the mutated positions for variants and 0 for all the other positions. This position encoding vector is then concatenated with the dot-product vector and used as the input of another fully connected neural network to predict the transcriptional activity. The prediction is normalized with a Sigmoid activation function so that the output value ranges from 0 to 1.

### 4.3. Training and Testing Rep2Mut

There are two steps to train Rep2Mut. First, we pretrained layers 1 and 2 (as shown in Figure 1) on another 37 protein datasets with various measurements of protein functions. The pretraining was used to optimize weights in the two neural networks. Afterward, we added layer 3 in Figure 1, and fine-tuned Rep2Mut to predict the GigaAssay activities. In both the pretraining and fine-tuning processes, we used the Adam optimizer [19] and MSE loss function in back-propagation. MSE is defined in Equation (1) where *n* is the number of data points, *Y_i_* is the observed activities, and *Ŷ* is the predicted activities.
(1)$$MSE=\frac{1}{n}\;\overset{n}{\underset{i=1}{\sum }}{\left(Y_{i}-{\hat{Y}}_{i}\;\right)}^{2}$$

During the fine-tuning process, we used a batch size of 8, and a learning rate of 1 × 10^−3^ for layer 3. Since layers 1 and 2 were optimized during the pretraining step, we used a smaller rate, and the learning rate for layers 1 and 2 was 1 × 10^−5^.

To compare Rep2Mut with the other methods, we used 10-fold cross validation. We randomly split the GigaAssay data into 10 groups, each of which had 10% of the experimental data. Each time, a group was used for testing and the remaining data for training. We repeated this process ten times and obtained the mean of the performance to evaluate Rep2Mut.

To test the performance of Rep2Mut with different sizes of training data, the variants in the dataset were shuffled and then split into two sets called the training (90%) and test (10%) set. Rep2Mut was learned on training data and evaluated on test data using the Pearson and Spearman’s correlation coefficients defined below. To avoid random splits, the process above was repeated 50 times, and the averaged performance was calculated for a final evaluation.

### 4.4. Evaluation Measurements

We used the Pearson and Spearman’s correlation coefficients to measure the performance of each tested method. We used the python package scipy to calculate both the Pearson and Spearman’s correlations for the prediction activities of a method. In detail, let *X* be the GigaAssay activities of a list of variants, and *Y* be the predicted activities of the same list, and then the Pearson correlation coefficients (PCC) are calculated using Equation (2), where *p* is the Pearson correlation coefficient, *x_i_* is the *i*th observed values in *X*, $\bar{x}$ is the mean of *X*, *y_i_* is the *i*th predicted values in *Y*, and *ȳ* is the mean of *Y*.
(2)$$p=\frac{\sum \left(x_{i}-\bar{x}\right)\left(y_{i}-\bar{y}\right)}{\sqrt{\sum {\left(x_{i}-\bar{x}\right)}^{2}\sum {\left(y_{i}-\bar{y}\right)}^{2}}\;}$$

Likewise, the Spearman’s rank correlation coefficients (sPCC) were estimated with Equation (3) where *sp* is the Spearman’s rank correlation coefficient; R(∗) is the ranking of items in ∗; cov(R(X), R(Y)) is the covariance of X and Y; $\sigma_{R\left(X\right)}$ is the standard deviation of X; $\sigma_{R\left(Y\right)}$ is the standard deviation of Y.
(3)$$sp=\frac{cov\left(R\left(X\right),\;R\left(Y\right)\right)}{\sigma_{R\left(X\right)}\sigma_{R\left(Y\right)}}$$

### 4.5. How to Use ESM to Predict Tat Variants’ Activities

ESM [9] has a diverse capability to estimate proteins’ activities and functions. Here, we used ESM (called ESM_pred so that it is different to the ESM released models) to estimate the GigaAssay activities that were not previously conducted. To determine the effect of the variants, the probability of each amino acid type at a position of interest is predicted in ESM_pred, and the variant effect is calculated based on the logarithmic ratio of the probability between the mutated amino acid and the WT amino acid in Equation (4), where *T* is the set of mutated positions; $x_{\backslash T}$ is the masked input sequence; $p\left(x_{t}=x_{t}^{mt}|x_{\backslash T}\right)$ is the probability assigned to the mutated amino acid $x_{t}^{mt}$; $p\left(x_{t}=x_{t}^{wt}|x_{\backslash T}\right)$ is the probability assigned to the wildtype.
(4)$$\underset{t\in T}{\sum }\log p\left(x_{t}=x_{t}^{mt}|x_{\backslash T}\right)-\log p\left(x_{t}=x_{t}^{wt}|x_{\backslash T}\right)\;$$

As recommended by ESM [9], five released ESM models (ESM-2 Public Release v1.0.3: esm1v_t33_650M_UR90S_1, esm1v_t33_650M_UR90S_2, esm1v_t33_650M_UR90S_3, esm1v_t33_650M_UR90S_4, and esm1v_t33_650M_UR90S_5) were used individually to predict the transcriptional effect after Tat variants. sPCC was then calculated for each model. In addition, the average prediction for the transcriptional effect of each Tat variant was determined by combining the predictions of five models, and estimated using sPCC.

### 4.6. How to Test DeepSequence on Tat Variants

DeepSequence [10] is a generative, unsupervised latent variable model to estimate the effects of the variants on biological sequences across a variety of datasets with deep mutational scanning. The model was learned in an unsupervised manner solely from sequence information, and grounded with biologically motivated priors, revealing a latent organization of sequence families. There are three steps to run DeepSequence. First, a protein sequence of interest was used as input of multiple sequence alignment (MSA) tools to generate multiple sequence alignments. We used the recommended tools by DeepSequence, EVcoupling from the website v2.evcouplings.org. DeepSequence [10] suggests the use of a bit score of 0.5 bits/residue as a threshold to generate MSA results. However, MSA results of the Tat protein with this score generated only 1.7 Seqs/L with 123 sequences, which was not enough to train DeepSequence. We thus tested two bit scores: 0.3 bits/residue with 1645 effective sequences and 23.8 Seqs/L as well as 0.25 bits/residue with 7871 effective sequences and 110.9 Seqs/L. Second, DeepSequence was trained with the sequences from MSA. Although DeepSequence is a generative model, each protein sequence requires a different model. We used MSA sequences with each bit score to retrain DeepSequence to generate separate models. Finally, the retrained models were used to predict the effect of the variants. However, both models predicted only ~10% (114) of the single variants, although 0.3 bits/residue produced better results.

### 4.7. The Framework of a Baseline Method

A simple baseline method was also designed and compared with Rep2Mut. This method uses the one-hot encoding of amino acids at each position as the input, and has three fully connected layers of feed forward networks: the input is a vector of 1720 elements, the first hidden layer generates a vector of 860 elements, and the second generates a vector of 256 elements. The output is the predicted activity for a variant. This method was trained with a batch size of 16 as well as a learning rate of 5 × 10^−4^, and tested with a similar strategy to Rep2Mut.

## 5. Conclusions

We designed a deep learning-based method that only used protein sequences to accurately predict the transcriptional activities of experimentally-determined Tat variants. With the representation learning from protein sequence models, our approach achieved a 0.94 Pearson correlation coefficient. This demonstrates that our deep learning-based method can precisely estimate the transcriptional activities of proteins with various variants and has great potential to be extended to complex mutations and other protein sequences. Although we used supervised learning while state-of-the-art methods such as ESM and DeepSequence models use unsupervised training, the superior performance makes our approach more promising for new applications. In particular, our method, trained on as little as 20 or 30% of data, was able to achieve a much better performance than the state-of-the-art methods, demonstrating its potential application on other proteins with limited training data. We plan to extend our methods to complex variant alleles and to other proteins for human disease studies.

## Figures and Tables

**Figure 1 ijms-24-06138-f001:**
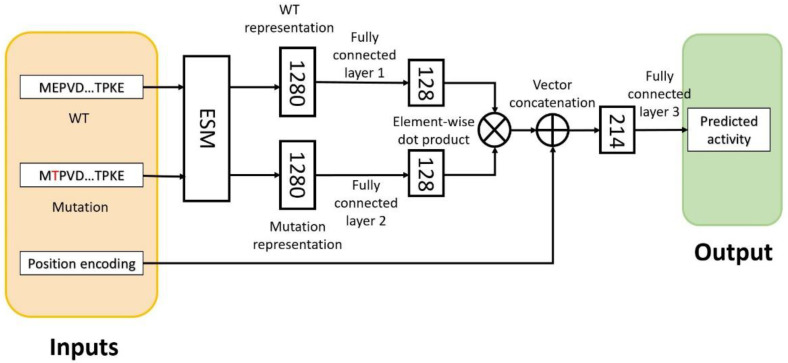
The generic architecture of the Rep2Mut model. Tat protein (86 amino acids) is shown as an example. An amino acid in red: a mutated residue; numbers in the rectangles indicate the size of the vectors; filled rectangles in brown: input; filled rectangles in green: output; cross symbol: elementwise dot product; plus symbol: concatenation of vectors.

**Figure 2 ijms-24-06138-f002:**
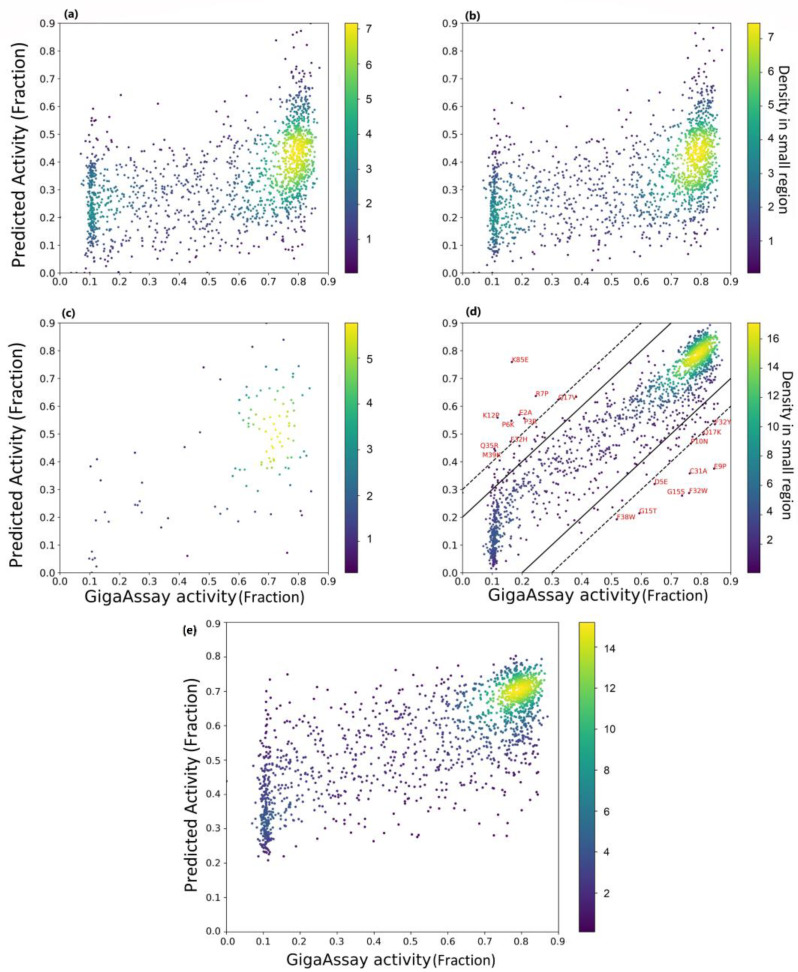
Comparison of activity estimation by Rep2Mut with two state-of-the-art methods. (**a**) ESM_pred: the best performance among the 5 ESM_pred estimation. (**b**) ESM_pred _avg: the performance of averaging the five ESM_pred estimation of variants. (**c**) DeepSequence. (**d**) Rep2Mut. The solid line: the error margins of 0.2, and the dashed line: an error margin of 0.3. Amino acid mutation outliers are labeled in red font. (**e**) A baseline method. Color legend at the right: the density of the dots in graph.

**Figure 3 ijms-24-06138-f003:**
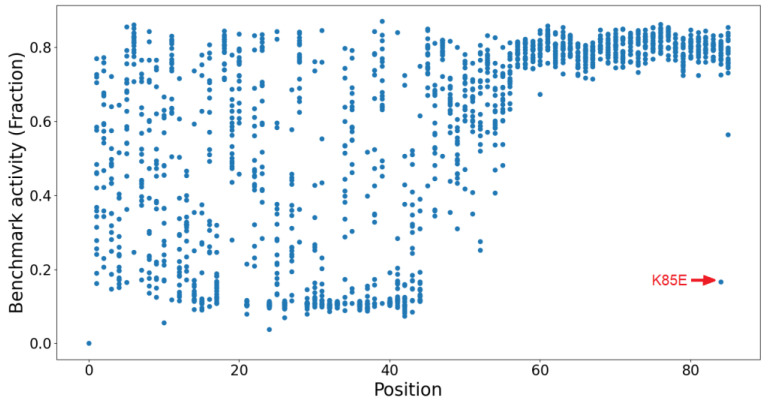
Tat variant activity is partially dependent upon the amino acid position. Each dot represents a Tat variant. The outlier K85E is highlighted in the figure.

**Figure 4 ijms-24-06138-f004:**
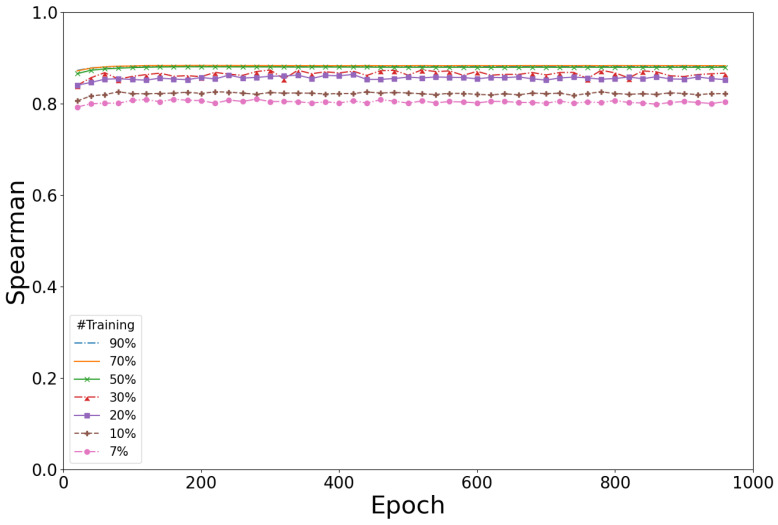
The sensitivity of Rep2Mut performance with the different numbers of training instances. X%: X% data are used to train Rep2Mut and (100-X)% for testing, and X is 90, 70, 50, 30, 20, 10, and 7 for different testing strategies. “#Training”: the numbers/percentages of training datasets.

**Figure 5 ijms-24-06138-f005:**
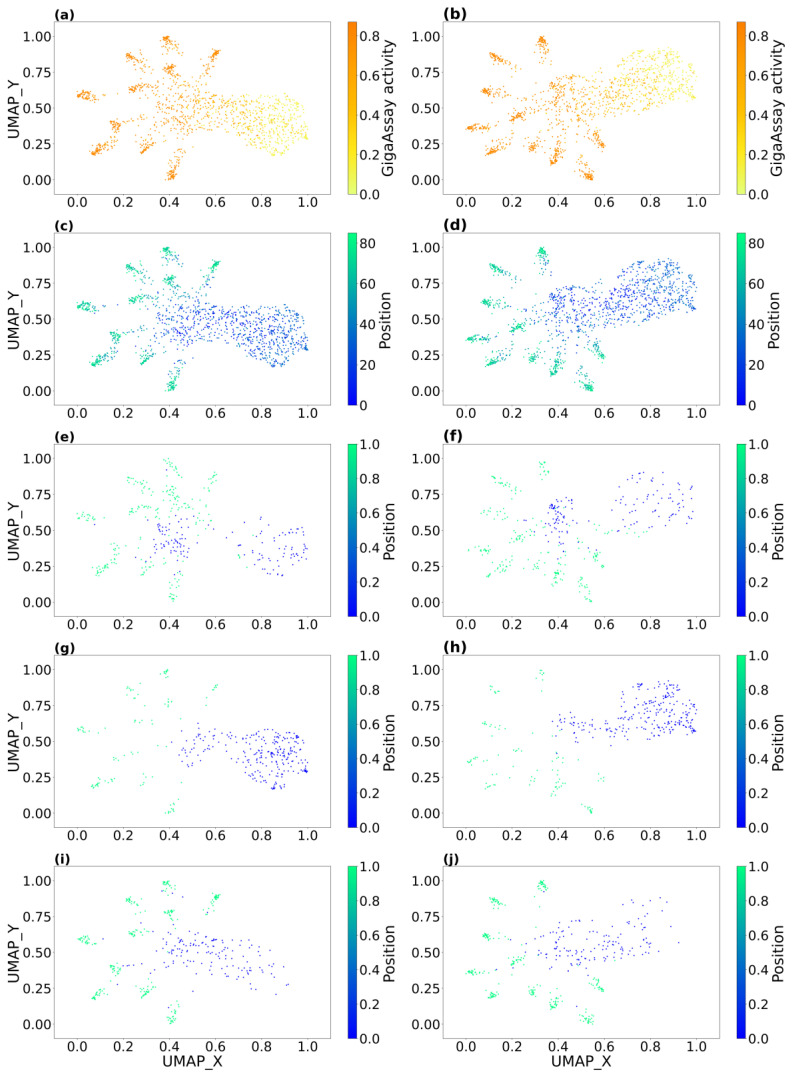
Visualization of the Rep2Mut final vectors after dimensionality reduction with UMAP: (**a**,**c**,**e**,**g**,**i**) with position vector; (**b**,**d**,**f**,**h**,**j**) without position vector; (**a**,**b**) colored by GigaAssay activities; (**c**–**j**):colored by positions; (**e**,**f**) positively charged amino acids (Arg, His, and Lys); (**g**,**h**) special cases of amino acids (Cys, Gly, and Pro); (**i**,**j**) polar uncharged amino acid (Ser, Thr, Asn, and Gln). In (**e**–**j**), 0: (blue) positions of variants lower than 45; 1: (green) positions of variants larger than 45, and this is why (**e**–**j**) have different color ranges from (**c**,**d**).

**Figure 6 ijms-24-06138-f006:**
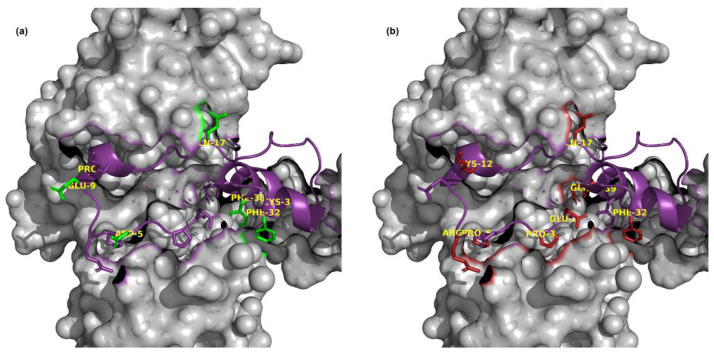
The structure (PDB ID: 4OR5) of the Tat: Cyclin T1 complex. Tat (purple cartoon) binds to Cyclin T1 (surface view). (**a**) Underestimated variants are colored green. (**b**) Overestimated variants are colored red.

**Table 1 ijms-24-06138-t001:** Pearson and Spearman correlation coefficients comparing the experimental activities to predictions from Rep2Mut and the state-of-the-art methods.

Prediction Method	Pearson	Spearman
ESM_pred	0.51	0.56
ESM_pred_avg	0.54	0.59
DeepSequence	0.57	0.41
The baseline method	0.76	0.72
Rep2Mut (wo_p ^1^)	0.91	0.87
Rep2Mut	**0.94**	**0.89**

^1^ “wo_p”: Rep2Mut without position encoding vector.

**Table 2 ijms-24-06138-t002:** Overestimated and underestimated outliers predicted by Rep2Mut.

				GigaAssay			Predicted		
Var	GA	Pred	#ID	Avg	Min	Max	Avg	Min	Max
Overestimation									
E2A	0.19	0.57	E2	0.44	P = 0.16; A = 0.19; C = 0.24	D = 0.77; T = 0.72; Q = 0.72	0.47	P = 0.28; I = 0.36; R = 0.36	S = 0.58; T = 0.57; Q = 0.57
P3R	0.21	0.56	P3	0.56	R = 0.20; K = 0.22; G = 0.34	L = 0.77; V = 0.75; I = 0.75	0.56	K = 0.41; D = 0.43; Y = 0.48	V = 0.64; S = 0.62; L = 0.62
P6K	0.16	0.55	P6	0.62	K = 0.16; R = 0.36; L = 0.45	W = 0.85; Y = 0.79; F = 0.77	0.61	R = 0.48; D = 0.51; E = 0.54	S = 0.67; H = 0.66; A = 0.66
R7P	0.25	0.64	R7	0.78	P = 0.24; K = 0.7; I = 0.75	E = 0.86; D = 0.85; S = 0.83	0.75	P = 0.63; W = 0.66; T = 0.69	E = 0.83; Q = 0.8; Y = 0.8
K12P	0.12	0.56	K12	0.70	P = 0.12; G = 0.50; T = 0.62	L = 0.83; Q = 0.82; N = 0.82	0.69	F = 0.55; P = 0.55; W = 0.59	Q = 0.8; A = 0.79; T = 0.78
Q17V	0.32	0.62	Q17	0.51	P = 0.11; W = 0.24; I = 0.24	K = 0.8; R = 0.78; A = 0.77	0.51	P = 0.32; F = 0.33; Y = 0.41	V = 0.62; M = 0.61; C = 0.59
F32H	0.16	0.47	F32	0.24	D = 0.09; K = 0.1; N = 0.1	Y = 0.84; W = 0.76; L = 0.55	0.27	G = 0.07; P = 0.11; E = 0.13	Y = 0.53; H = 0.47; M = 0.42
Q35R	0.11	0.44	Q35	0.42	K = 0.08; D = 0.1; R = 0.11	H = 0.79; M = 0.74; Y = 0.71	0.43	P = 0.26; G = 0.33; E = 0.33	H = 0.55; A = 0.55; M = 0.51
M39K	0.11	0.45	M39	0.45	W = 0.1; K = 0.1; R = 0.1	L = 0.84; I = 0.78; V = 0.78	0.45	P = 0.22; R = 0.28; D = 0.28	V = 0.77; I = 0.61; S = 0.6
K85E	0.17	0.76	K85	0.76	E = 0.16; W = 0.72; F = 0.75	V = 0.83; D = 0.81; Q = 0.81	0.78	P = 0.65; M = 0.73; W = 0.73	S = 0.87; T = 0.83; H = 0.82
Underestimation									
D5E	0.64	0.32	D5	0.28	F = 0.15; I = 0.16; R = 0.17	E = 0.64; S = 0.51; C = 0.4	0.32	I = 0.18; L = 0.19; M = 0.22	N = 0.46; S = 0.46; H = 0.41
E9P	0.84	0.37	E9	0.45	W = 0.13; F = 0.15; Y = 0.17	P = 0.84; A = 0.81; D = 0.76	0.45	F = 0.25; W = 0.32; R = 0.32	D = 0.66; A = 0.61; Q = 0.58
P10N	0.77	0.46	P10	0.43	W = 0.12; F = 0.14; M = 0.17	N = 0.76; A = 0.74; S = 0.74	0.43	W = 0.29; D = 0.32; Y = 0.33	A = 0.62; S = 0.57; C = 0.53
G15T	0.59	0.21	G15	0.21	E = 0.09; F = 0.11; I = 0.11	S = 0.73; T = 0.59; P = 0.35	0.23	Y = 0.11; I = 0.14; F = 0.16	Q = 0.36; M = 0.33; V = 0.33
G15S	0.74	0.28	G15	0.21	E = 0.09; F = 0.11; I = 0.11	S = 0.73; T = 0.59; P = 0.35	0.23	Y = 0.11; I = 0.14; F = 0.16	Q = 0.36; M = 0.33; V = 0.33
Q17K	0.81	0.50	Q17	0.51	P = 0.11; W = 0.24; I = 0.24	K = 0.8; R = 0.78; A = 0.77	0.51	P = 0.32; F = 0.33; Y = 0.41	V = 0.62; M = 0.61; C = 0.59
C31A	0.76	0.36	C31	0.24	P = 0.09; Y = 0.1; E = 0.1	A = 0.76; S = 0.73; V = 0.42	0.25	R = 0.11; K = 0.14; D = 0.14	S = 0.53; V = 0.51; T = 0.49
F32Y	0.85	0.53	F32	0.24	D = 0.09; K = 0.1; N = 0.1	Y = 0.84; W = 0.76; L = 0.55	0.27	G = 0.07; P = 0.11; E = 0.13	Y = 0.53; H = 0.47; M = 0.42
F32W	0.76	0.29	F32	0.24	D = 0.09; K = 0.1; N = 0.1	Y = 0.84; W = 0.76; L = 0.55	0.27	G = 0.07; P = 0.11; E = 0.13	Y = 0.53; H = 0.47; M = 0.42
F38W	0.52	0.19	F38	0.15	D = 0.08; Q = 0.09; R = 0.09	W = 0.51; Y = 0.39; L = 0.15	0.14	R = 0.06; K = 0.07; S = 0.07	I = 0.22; V = 0.22; W = 0.19

Var: the variants, GA: GigaAssay, Pred: predicted activities, #ID: WT amino acids with variant positions, Min: the 3 minimum GigaAssay or predicted values for that variant, Max: the 3 maximum GigaAssay or predicted values for that variant. “X = Y”: X is the mutated amino acid type, and Y is the activities.

## Data Availability

The data and the scripts for data analysis are publicly available at https://github.com/qgenlab/Rep2Mut.

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
