# Peer review of "Accurate Prediction of Transcriptional Activity of Single Missense Variants in HIV Tat with Deep Learning"

_ijms, 2023, doi:10.3390/ijms24076138_

Round 1

Reviewer 1 Report

Brief Summary

Derbel et al. embark on a journey to study the transcription of HIV virus within the human cell. Specifically, they focus on predicting how HIV transcription is affected by Tat single missense substitution variants. Derbel et al. combine GigaAssay transcriptional activity data with computational deep learning algorithm to predict the activity of Tat variants. Furthermore, this approach can be implemented in the future to more complex sequence variations of Tat and to other viral transcription factors. 

Significance

The research of the HIV virus has turned HIV infection from a deadly disease to a chronic lifelong infection. Yet, the rise in antiviral resistance, adverse effects from lifelong treatment, and drug-drug interactions threaten to limit the success of available HIV treatments and creates an urgent need for new drugs. Understanding the molecular biology of the HIV virus is critical to the development of new treatments for HIV, specifically its replication mechanism. On the one hand, the high sequence variability as a result of its replication mechanism lends HIV its immune and drug evasiveness. On the other hand, the sequence variability nature can serve as a new drug target when better understood. 

Recommendations: 

I recommend accepting this paper with minor revisions for publication at the IJMS Journal. I am listing below minor suggestions for clarifying details described in this commentary. 

Comments: 

1.     Reference technical errors throughout the text. Line 74, 83, 86, 88… 

2.     Please add references to the result section of “Effect of amino acid position on activity prediction”, about the C-terminal of Tat protein. In addition, if K85E is present in figure 3, please highlight it in the figure using arrow/circle/color, and add it to the figure legend as well. 

3.     Figure 5, please double check the axis titles, color assignment, and legend description. Why a-d colored yellow to orange, while e-j colored blue to red? Why is position ranging 1-86 in some panels, and 0-1 in other panels? Legend says panels c-j colored by activity, while the c-j panels present a position color range? 

4.     Figure 6. The highlighted amino acids of Tat are hard to see over the bright cyan surface of the Cyclin T1. Please choose a different color scheme. For example, use another color for the Cycline T1, wither pale light blue, or light gray.

5.     Line 249: “The observation of the erroneous predictions at the Cyclin T1 interface suggests that Cylcin T1 accessibility or interactions may differ in the cell lines used in the GigaAssay“. This statement is too strong. I suggest changing it to “One explanation to the observation of the erroneous predictions at the Cyclin T1 interface could suggests that Cylcin T1 accessibility or interactions may differ in the cell lines used in the GigaAssay. Another explanation could be that our prediction does not catch the structural complexity of the protein-protein interaction in this area. A third explanation could be that these amino acids are important for another interaction. “ 

Author Response

Comments from reviewer 1:

Recommendations:
I recommend accepting this paper with minor revisions for publication at the IJMS Journal. I am listing below minor suggestions for clarifying details described in this commentary.

Authors’ response: Thank you very much for your wonderful comments. We have revised the manuscript according to your suggestions.

Comments: 

  1. Reference technical errors throughout the text. Line 74, 83, 86, 88… 

Authors’ response: Thank you for pointing this out and our apologies for this. This happened after uploading doc file, since the references in our local doc file were correct. To avoid this, we will upload both PDF and doc files and the PDF manuscript has no reference error.

  1. Please add references to the result section of “Effect of amino acid position on activity prediction”, about the C-terminal of Tat protein. In addition, if K85E is present in figure 3, please highlight it in the figure using arrow/circle/color, and add it to the figure legend as well. 

Authors’ response: Thank you for your comment: we have added a reference to the section of the effect of amino acid position on activity prediction. We have also highlighted the mutation K85E in Figure 3. Please refer to Figure 3 in Page 5.

  1. Figure 5, please double check the axis titles, color assignment, and legend description. Why a-d colored yellow to orange, while e-j colored blue to red? Why is position ranging 1-86 in some panels, and 0-1 in other panels? Legend says panels c-j colored by activity, while the c-j panels present a position color range?

Authors’ response: Thank you for this great comment. We have regenerated the plot in Figure 5 (Page 8) so that the color assignment is consistent: Figure 5 a and b are colored by the activity with a color range from yellow to orange and Figure 5 c-j are colored by the position with a color range from blue to green. For Figure e-j, blue represents positions lower than 45 and green represents positions greater than 45. This is why Figure 5 c-d has a different value range from Figure 5 e-j.

  1. Figure 6. The highlighted amino acids of Tat are hard to see over the bright cyan surface of the Cyclin T1. Please choose a different color scheme. For example, use another color for the Cyclin T1, wither pale light blue, or light gray.

Authors’ response: Thank you for pointing this out. We have regenerated Figure 6 (Page 9) according to your suggestion, and the labels are clearer in the revised figure.

  1. Line 249: “The observation of the erroneous predictions at the Cyclin T1 interface suggests that Cyclin T1 accessibility or interactions may differ in the cell lines used in the GigaAssay“. This statement is too strong. I suggest changing it to “One explanation tothe observation of the erroneous predictions at the Cyclin T1 interface could suggests that Cyclin T1 accessibility or interactions may differ in the cell lines used in the GigaAssay. Another explanation could be that our prediction does not catch the structural complexity of the protein-protein interaction in this area. A third explanation could be that these amino acids are important for another interaction. “ 

Authors’ response: Thank you for raising our awareness to the alternative explanations. We have incorporated these suggestions in the revised manuscript (Page 9).

Reviewer 2 Report

I had a great opportunity to review manuscript entitled: 'Accurate prediction of transcriptional activity of single missense variants of HIV Tat using deep learning' which is considered for publication in IJMS. The manuscript is elaborated on an interesting topic and clearly summarize new data valuable for the research community. The author has done a good job at describing the problem, the methods and the results.

GENERAL COMMENTS:
TITLE
The paper title is well stated, it is informative and concise.

ABSTRACT, INTRODUCTION
Abstract is well written with the key findings of the study. Introduction is concise, focused and informative.

MATERIAL AND METHODS
Material and research methods are presented appropriately and clearly. Experimental setup and the description in the methods section are well structured, and the computational analysis is done alright.

RESULTS
The results obtained in this study are interesting. Results presented correctly.

DISCUSSION
In general, the discussion of results is correct.

LITERATURE
The items of literature included in the paper are rather sufficient and adequate to the subject of the paper.

Author Response

Comments from reviewer 2:

Thank you for your time and effort to review our manuscript and we appreciate the generally positive review.

Reviewer 3 Report

The authors proposed a deep learning method, called Rep2Mut, to predict the transcriptional activity of missense variants. Rep2Mut takes three types of input, representations of wild-type (WT) sequences and mutated sequences, and mutation position encoding vector. Rep2Mut achieves a 0.94 Pearson correlation coefficient when compared with the experimental data generated by GigaAssay, outperforming the recent methods, the evolutionary scale modelling (ESM) protein language model and DeepSequence. It is interesting to know that mutation position encoding contributes to the final prediction model. The paper is well written, and the experiment results are promising. I only have a few minor suggestions.

1.      The 15th reference is not complete, missing several fields, such as journal names, published data and so on.

2.      The Figures are not cited well. It shows “Error! Reference source not found.” in the manuscript.

3.       Line 274, duplicated citations in “ESM [9], [14], [14] is a …”

Author Response

Comments from reviewer 3:

  1. The 15threference is not complete, missing several fields, such as journal names, published data and so on.

Authors’ response: Thank you very much for pointing this out. We have added all the missing information for the 15th reference (Page 13): Devlin J, Chang M-W, Lee K, Toutanova K. BERT: Pre-training of Deep Bidirectional Transformers for Language Understanding. Proceedings of NAACL-HLT. 2019;4171–86.

  1. The Figures are not cited well. It shows “Error! Reference source not found.” in the manuscript.

Authors’ response: Thank you very much for pointing this out. This error happened after we uploaded the doc file, since our local doc file does not have this error. To avoid this type of error, we uploaded both PDF and doc file, and these types of errors are not in the PDF version.

  1. Line 274, duplicated citations in “ESM [9], [14], [14] is a …”

Authors’ response: Thank you for pointing this out. We have removed the duplicated citation (Page 10) and double checked all other references.

Reviewer 4 Report

­­The proposed manuscript is devoted to a study on a deep-learning methodology for estimation of the transcriptional activity of missense variants in HIV Tat using protein sequences .

Preliminaries to the research area are provided. Information related to the development of AIDS and its consequences as well as the role of Tat as an essential regulatory gene involved in HIV genome transcription and replication are described in detail. Previous related approaches are overviewed.  

The proposed methodology and the obtained results are carefully described. The properties of the method are evaluated showing that it is able to achieve better performance in comparison with other approaches used in this field.

The presentation of the main results is clear and comprehensive. From a formal point of view, all the contents seems to be correct. The results are valuable and worthy of being published taking into account their possible applications in clinical practice, in particular for better understanding of AIDS pathology and treatment, especially for investigation of other proteins with limited training data.

Minor revisions are suggested to improve the quality of the exposition:

p. 2, line 69: I suggest the aim and the novelty of the study to be better explained in the introductory section.

p. 2, line 74 and later: There are many errors of this type: “ Error! Reference source not found”.

p. 9, line 257: I propose the Section ”Materials and Methods” to be moved before the Results and Discussion.

p. 12-13: The formatting of the references is different, e.g. Ref.2 from the others, it should be unified in accordance with the requirements of the Journal.

Author Response

Comments from reviewer 4:

p.2line 69: I suggest the aim and the novelty of the study to be better explained in the introductory section.

Authors’ response: Thank you for your comment. We have revised the last paragraph in the introduction to better explain the aim and novelty of our work (Page 2).

p.2, line 74 and later: There are many errors of this type: “ Error! Reference source not found”.

Authors’ response: Thank you very much for pointing this out. The reference errors happened after uploading the doc file. To avoid reference errors, we uploaded both PDF and doc file, and the PDF file is free of these reference errors.

p.9line 257: I propose the Section ”Materials and Methods” to be moved before the Results and Discussion.

Authors’ response: Thank you for your suggestion, and we are happy to change the order of sections. However, based on the instructions of the journal (https://www.mdpi.com/journal/ijms/instructions), the required section order is “Research manuscript sections: Introduction, Results, Discussion, Materials and Methods, Conclusions (optional).

p.12-13: The formatting of the references is different, e.g. Ref.2 from the others, it should be unified in accordance with the requirements of the Journal.

Authors’ response: Thank you for pointing this out. We have revised the references (Page 13) in accordance with your comment and the requirements of the journal.